# Distributed Environments for Ocean Forecasting: the role of Cloud Computing

Stefania Ciliberti[1] and Gianpaolo Coro[2]

[1]Nologin Oceanic Weather Systems, , Spain
[2]Istituto di Scienza e Tecnologie dell'Informazione "Alessandro Faedo", Area della Ricerca CNR di Pisa, Pisa, Italy

*Correspondence to*: stefania.ciliberti@nowsystems.eu

**Abstract.** Cloud computing offers an opportunity to innovate traditional methods for provisioning of scalable and measurable computed resources as needed by operational forecasting systems. It offers solutions for more flexible and adaptable computing architecture, for developing and running models, for managing and disseminating data to finally deploy services and applications. The review discussed on the key characteristic of cloud computing related on on-demand self-service, network access, resource pooling, elasticity and measured services. Additionally, it provides an overview of existing service models and deployments methods (e.g., private cloud, public cloud, community cloud, and hybrid cloud). A series of examples from the weather and ocean community is also briefly outlined, demonstrating how specific tasks can be mapped on specific cloud patterns and which methods are needed to be implemented depending on the specific adopted service model.

## 1 Introduction

Cloud computing presents an opportunity to rethink traditional approaches used in operational oceanography (Vance et al, 2016), since it can enable a more flexible and adaptable computing architecture for observations and predictions, offering new ways for scientists to observe and predict the state of the ocean and, consequently, to build innovative downstream services for end-users and policy makers. Operational Ocean Forecasting Systems (OOFS) are sustained by a solid backbone composed of satellite and marine observation networks for Earth observations (i.e., data) and state-of-the-art numerical models (i.e., tools) that deliver products according to agreed standards (i.e., ocean predictions, indicators, etc.): the workflow is well represented by the ocean value chain, as described in Bahurel et al. (2010) and Alvarez-Fanjul et al. (2022). OOFS massively use high performance computing (HPC) to process data and run tools, whose results are shared and validated according to agreed data standards and methodologies, that can result in a remarkable computational cost, not always affordable for research institutes and organizations. Additionally, when building services, it is also important to guarantee lower-latency, cost-efficiency and scalability, together with reliability and efficiency. In such framework, cloud computing can represent an opportunity for expanding the capabilities of forecasting centres in managing, producing, processing and sharing ocean data. It implies adopting, evolving and sustaining standards and best practices to enhance management of ocean value chain, to optimize the

OOFS processes and to allow rationalization of requirements and specifications to properly account for operating a forecasting system (Pearlman et al., 2019).

Cloud technology has been dramatically evolved in the last decades: private sector has extensively used cloud computing for enabling scalability and security, leveraging it for Artificial Intelligence (AI) and Machine Learning (ML) framework, Internet of Things (IoT) integration and HPC to optimize and innovate operations. It plays also a crucial role in enhancing data interoperability and FAIR (findable, accessible, interoperable and reusable, Wilkinson et al., 2016) principles, through standardization of formats, APIs and access protocols, ensuring that datasets can be easily shared, accessed, and reused by researchers globally.

Considering OOFS, the computational and programming models offered by cloud computing can largely support real time data processing, scalable model runs, data sharing and elastic operations, facilitating the integration of AI/ML techniques (Heimbach et al., 2025) and the development of applications for Blue Economy and society (Veitch et al., 2024) in operational frameworks. More in detail, cloud computing can provide a powerful and collaborative platform for the development and running of operational models, for management and dissemination of data, for building and deploying services to downstream business and applications, and finally for analyses and visualization of oceanographic products, enabling researchers to tackle larger and more complex problems without the burden of building and maintaining computing and storage infrastructures. However, challenges such as data transfer latency, security and potential vendor lock-in must be addressed, including the high-costs for running complex modelling systems.

This paper explores today capabilities in cloud computing technology with an outlook on benefit and challenges in adopting this paradigm in OOFS. The reminder of this paper is organized as follows: Section 2 presents cloud computing foundational key concepts, highlighting some existing initiatives from the private sectors; Section 3 discusses on opportunities and challenges for ocean prediction in adopting cloud technologies, presenting existing international initiatives worldwide as examples. Section 4 concludes this paper.

## 2 Key concepts of Cloud Computing

### 2.1 A brief history of Cloud Computing

Cloud computing is a specialized form of distributed computing that introduces utilization models for remotely provisioning scalable and measured computing resources (e.g., networks, servers, storage, applications, and services) (Mahmood et al., 2013), offering organizations different benefits for their business services and applications: scalability, cost savings, flexibility and agility, reliability and availability, collaboration and accessibility, innovation and experimentation, and sustainability.

The term originated as a metaphor for the Internet which is, in essence, a network of networks providing remote access to a set of decentralized IT resources. In the early 1960s, J. McCarthy introduced the concept of computing as Utility: "If computers of the kind I have advocated become the computers of the future, then computing may someday be organized as a public utility

just as the telephone system is a public utility.… The computer utility could become the basis of a new and important industry". This idea opened to the concept of having services on the Internet so users could benefit of them for their applications. In the same period, J. C. R. Licklider envisioned a world where interconnected systems of computers could communicate and inter-operate: that was the milestone of the modern cloud computing. In the late 1990s, R. Chellappa introduced for the first time the term "cloud computing" as a new computing paradigm (Chellappa, 1997), "where the boundaries of computing will be determined by economic rationale rather than technical limits alone", dealing with concepts such as expandable and allocatable resources that can ensure cost-efficiency, scalability, and business value. In the same period, Compaq Computer Corporation adopted the concept of "cloud" in its business plan, as term for evolving the technological capacity of the company itself in offering new scalable and expandable services to customers over the Internet. The last 2 decades have been characterized by a rapid expansion of Cloud Computing: while the general public has been leveraging forms of Internet-based computer utilities since the mid-1990s as form of search engines, e-mail services, social media platforms, etc., it wasn't until 2006 that the term cloud computing emerged, when Amazon launched its Simple Storage Service (Amazon S3) followed by the Elastic Compute Cloud (Amazon EC2) service, enabling organizations to lease computing capacity and storage to run their business applications. In 2008, Google launched the Google App Engine, a cloud computing platform used as a service for developing and hosting web applications; then, in 2010 Microsoft launched Azure as a cloud computing platform and service provider that provides scalable, on-demand resources to customers to build applications globally; in 2012, Google launched the Google Compute Engine which enables users to launch virtual machines (VM) on demand

To understand the framework over which cloud computing is built, it is fundamental to refer to standards and best practices provided by the North American National Institute for Standard and Technology (NIST) (Mell and Grance, 2011): "cloud computing is a model for enabling ubiquitous, convenient, on-demand network access to a shared pool of configurable computing resources that can be rapidly provisioned and released with minimal management effort or service provider interaction".NIST further elaborates on cloud computing providing a Cloud Computing Reference Architecture based on five Essential Characteristics, three Service Models, and four Deployment Models.

**2.2 An outlook to NIST definitions**

Cloud computing **Essential Characteristics** defined by NIST can be considered as reference guidelines for both providers and clients to ensure scalable, cost-effective and accessible resources to fit specific needs. Table 1 shows a summary of the Essential Characteristics' definitions as provided in Mell and Grance (2011), offering the client and provider's perspectives with some examples that show how cloud solutions ensure scalability, flexibility and efficiency

**Table 1: NIST Cloud Computing Essential Characteristics: client/provider perspectives and examples**

| Characteristics | Primary Focus | Client Perspective | Cloud Provider Perspective | Example |
|---|---|---|---|---|
| | | | | |

| | | | |
|---|---|---|---|
| On-Demand Self-Service | Users can provision computing resources (e.g., storage, VMs) automatically, without requiring human interaction with the service provider. | Users can request and configure resources like virtual machines, storage, or applications when needed, directly from a web interface or API. | Automatically provide resources in response to user requests without manual intervention. | A developer launches a virtual machine on a cloud platform using a dashboard or API in minutes, without needing to contact support. |
| Broad Network Access | Cloud resources are available over a network and accessible through standard mechanisms from various devices | Users can access cloud services from a range of devices (e.g., mobiles, PCs, etc.) through standard protocols like HTTP/HTTPS and APIs. | Ensure cloud services can be accessed consistently and securely from different client devices. | A user edits a document stored in the cloud from a laptop at home, and then continues editing from a smartphone while commuting. |
| Resource Pooling | Cloud providers pool resources to serve multiple users (tenants) dynamically, with no fixed assignment to any one user. | Users don't know the exact physical location of the resources they are using, but they get what they need as required. | Dynamically allocate physical and virtual resources across many customers to maximize efficiency and utilization. | Multiple customers use the same set of servers and storage, but their workloads are isolated through virtualization technologies for security. |
| Rapid Elasticity | Cloud resources can be quickly scaled up or down to meet demand, often appearing limitless to the user. | Users can automatically scale their resources up or down based on their needs, without delays. | Automatically add or remove resources in response to changing demand, ensuring that the user has sufficient capacity. | An e-commerce website automatically scales up its computing resources during a flash sale, then scales down when the traffic subsides. |

| | | | | |
|---|---|---|---|---|
| Measured Service | Cloud systems automatically control and optimize resource usage by tracking it and charging based on actual consumption. | Users only pay for the amount of resources (e.g., storage, CPU, bandwidth) they actually use, with detailed reporting. | Track resource consumption at various levels (e.g., storage, CPU usage) and optimize based on real-time monitoring. | A company receives a monthly bill detailing how much computing power and storage they used, ensuring that they are billed accurately based on consumption. |

NIST specifies three possible cloud **Services Models**: Infrastructure as a Service (IaaS), Platform as a Service (PaaS) and Software as a Service (SaaS). They define the foundational cloud services' characteristics clients need, to ensure adequate levels of management, flexibility and control. Table 2 presents Service Models' definitions as provided in Mell and Grance (2011), discussing examples where they are used.

**Table 2: NIST Cloud Computing Service Models.**

| Service Model | Primary Focus (from Mell and Grance, 2011) | Client Perspective | Provider Perspective | Use Cases |
|---|---|---|---|---|
| Infrastructure as a Service (IaaS) | The capability provided to the consumer is to provision processing, storage, networks, and other fundamental computing resources where the consumer can deploy and run arbitrary software. | Renting and managing computing resources in a virtualized infrastructure. | Provisioning of computing resources in a virtualized infrastructure. | Suitable for organizations that want full control over their infrastructure resources (virtual machines, networks, storage) that want their flexibility in customizing software stack and applications, including data processing and backup. Examples: Amazon EC2, Microsoft Azure, etc. |
| Platform as a Service (PaaS) | The capability provided to the consumer is to deploy onto the cloud infrastructure consumer- | Easing applications deployment without taking care of the infrastructure and | Provisioning and management of the platform. | Suitable for developers and organizations that want to develop, deploy and maintain |

| | | | |
|---|---|---|---|
| | created or acquired applications created using programming languages, libraries, services, and tools supported by the provider. | middleware. Dependency on provider's platform. | | applications without the burden of managing the underlying infrastructure (virtual machines, network and storage), that is provisioned and deployed by the providers. Examples: Google App Engine, Microsoft Azure App Services, etc. |
| *Software as a Service (SaaS)* | The capability provided to the consumer is to use the provider's applications running on a cloud infrastructure. | Using directly software applications via Internet (e.g., web browser or using a client), decreasing costs related to licences. | Provisioning and management of the software applications, including customer support. | It enables organizations to focus on their core business activities while relying on the expertise and infrastructure provided by the SaaS provider. Examples: Google Drive, Dropbox, Microsoft 365. |

Beside the NIST definitions, similar to PaaS another service model is the Serverless model (or Function as a Service - FaaS),
that is the capability provided to the user to abstract infrastructure concerns away from applications, where developers can
implement application functionality as invokable functions/services whilst providers automatically provision, deploy, and
scale these services based on a range of criteria, including efSantiago de Compostelaficiency, cost, load balancing, etc.
Examples of Serverless/FaaS services are AWS Lambda[1] and Fargate[2], Microsoft Azure Functions[3], Google Cloud Functions[4],
Scaleway Serverless Functions[5].

Cloud computing deployment models can be based on different approaches, offering organizations options for workload
placement, application development, and resource allocation to optimize their cloud strategy based on their needs, cost

[1] https://aws.amazon.com/lambda
[2] https://aws.amazon.com/fargate
[3] https://azure.microsoft.com/en-us/products/functions
[4] https://cloud.google.com/functions
[5] https://www.scaleway.com/en/serverless-functions

considerations, performance requirements, compliance regulations and desired level of control. The four cloud computing **Deployment Models** identified by NIST are reported in Table 3 with a description and some examples.

**Table 3: NIST Cloud Computing Deployment Models.**

| Deployment Model | Description | Examples |
|---|---|---|
| Private Cloud | Deployment of cloud infrastructure and services exclusively for a single organization or entity. In a private cloud, the computing resources, such as servers, storage, networking, and virtualization technologies, are dedicated to and managed by the organization itself. The infrastructure can be hosted on-premises within the organization's own data centers or in a dedicated off-site facility. | Open-source software solutions such as CloudStack[6], OpenNebula[7], Openstack[8], allow organizations to build their own private cloud computing solutions. |
| Public Cloud | Use of cloud services provided by third-party vendors over the internet. The infrastructure and resources in the public cloud are shared among multiple customers and the cloud service provider is responsible for managing and maintaining the underlying hardware, software, and infrastructure. Users can access and utilize the services on a pay-as-you-go basis, typically through a subscription or usage-based pricing model. | Examples of Public Cloud providers are Alibaba[9], Amazon Web Services[10], Google Cloud Platform[11], Hetzner[12], Microsoft Azure[13], Scaleway[14]. |

---

[6] https://cloudstack.apache.org
[7] https://opennebula.io
[8] https://www.openstack.org
[9] https://www.alibabacloud.com
[10] https://aws.amazon.com
[11] https://cloud.google.com
[12] https://www.hetzner.com/cloud
[13] https://azure.microsoft.com
[14] https://www.scaleway.com/en

| | | |
|---|---|---|
| *Community Cloud* | Cloud infrastructure and resources are shared among organizations with common interests, such as industry-specific regulations, security requirements, or collaborative projects. In a community cloud, the infrastructure is designed and managed for the specific needs of the community members, and it allows organizations within the community to share costs, resources, and expertise while maintaining a higher level of control and customization compared to public cloud services. | EGI [15] is a federation of different European Data Centers providing a cloud infrastructure for research communities. The European Open Science Cloud (EOSC[16]) is an environment for hosting and processing research data to support EU science, built on top of EGI cloud infrastructure. The European Weather Cloud [17] will deliver data access and cloud-based processing capabilities for the European Meteorological Infrastructure (EMI) and their users. The D4Science[18] e-infrastructure (Assante et al., 2019) is the core of the Blue-Cloud[19] Virtual Research Environments (VREs): it implements proven solutions for connecting to external services and orchestrates distributed services, which will be instrumental for smart connections to other e-infrastructures in Blue-Cloud, including EUDAT and DIAS (WekEO). |
| *Hybrid Cloud* | It combines both public and private cloud environments to create a unified computing infrastructure, allowing organizations to host some applications or data in a private cloud (i.e. greater control, security and compliance), while utilizing public cloud services for other applications or workloads (i.e. scalability, cost-effectiveness and flexibility for workload burst/on-demand peaks). The hybrid approach provides the ability to address specific requirements, such as regulatory compliance or data sovereignty, by keeping sensitive data within | Netflix[20] uses a hybrid cloud storage solution in order to store and move assets across Amazon AWS S3 and multiple on-premises storage systems. |

---

[15] https://www.egi.eu
[16] https://eosc.eu
[17] https://www.europeanweather.cloud
[18] https://www.d4science.org/
[19] https://www.blue-cloud.org/e-infrastructures/d4science
[20] https://aws.amazon.com/solutions/case-studies/netflix-storage-reinvent22

| | a private infrastructure while utilizing the public cloud for less sensitive workloads. | |
|---|---|---|

Beside the cloud deployment models identified by NIST, there are few other approaches that are worth mentioning that provide further capabilities to the organizations that decide to embrace cloud technology.

**Multi-cloud computing** refers to the strategy of using multiple cloud service providers, allowing organizations leveraging the services of two or more public/private cloud providers or a combination public-private, combining their offerings to build and manage their applications and infrastructure. This approach allows businesses to take advantage of the strengths and capabilities of different cloud providers, such as cost-effectiveness, performance, geographic coverage, or specialized services. It also offers increased flexibility, redundancy, and mitigates the risk of vendor lock-in (Hong et al., 2019). Multi-cloud solutions, that can be based on open-source technologies such as Kubernetes, offer the possibility to ease migration of applications, improving portability since they support containerization and microservices. Major challenges include the complexity in the management of the infrastructure, issues with integration and interoperability and security. The **edge-computing** paradigm enables data analysing, storage and offloading computations near the edge devices (such as Internet of Things – IoT – devices, sensors, mobile devices, etc.) to improve response time and save bandwidth (Pushpa and Kalyani, 2020). This approach aims at minimizing the data volume to process in the cloud, reducing network costs and bandwidth utilization and increasing reliability and scalability. Major challenges include the complexity in the management of the edge devices, security potentially affected by devices' vulnerability and synchronization of communications between edge devices and cloud infrastructure.

**Distributed cloud-edge computing**, one of the main innovation streams for cloud computing, combines elements of cloud computing with edge computing, extending the capabilities of the traditional centralized cloud infrastructure by distributing cloud services closer to the edge of the network, where data is generated and consumed, rather than relying solely on centralized data centres. By moving cloud services closer to where data is generated, latency (defined as the delay in network communication) is reduced, allowing fast response times, and real-time or time-sensitive applications (e.g., collection of observations from automated sensors and systems for guaranteeing efficiency in operations; early warning systems for disaster management and safety) can benefit from faster response times and improved performance. This is especially crucial for applications requiring immediate data processing and low latency. Recently, public cloud providers started to offer pre-configured appliances (e.g. AWS Outpost, Azure Stack) that brings the power of the public cloud to the private and edge cloud and have defined collaborations with telcos (e.g. AWS and Vodafone, Google and ATT) to create 5G edge services. Furthermore, the main open source cloud management platforms provide extensions (OpenNebula ONEedge, OpenStack StarlingX, Kubernetes KubeEdge) for enhancing private clouds with capabilities for automated provisioning of compute, storage and networking resources and/or orchestrate virtualized and containerized application on the edge. Major challenges

include ensuring data security across the distributed locations, for a safe communication between cloud and edge, and resource management and network reliability.

Based on NIST's definitions as discussed before, Table 4 summarizes how the five Essential Characteristics apply across the four Deployment Models (Public, Private, Hybrid, and Community Cloud) to support the selection of the right cloud model with respect to efficiency in costs and performances, security and management.

**Table 4: Mapping Essential Characteristics on type of cloud Deployment Models.**

| Essential Characteristic | Deployment Model | | | |
|---|---|---|---|---|
| | **Private Cloud** | **Public Cloud** | **Community Cloud** | **Hybrid Cloud** |
| On-Demand Self-Service | Managed internally, self-service for internal teams | Users provision services via public provider's API or portal | Self-service for community members, often through secure portals | Self-service across both public and private clouds, with potential for complex management |
| Broad Network Access | Limited to internal users or authorized external users (VPN, private network) | Accessible over the public internet via standard protocols (e.g., HTTP) | Restricted to community members with specific access | Accessible over both public and private networks, often with encrypted or dedicated connections |
| Resource Pooling | Resources are pooled internally for organizational needs | Resources are pooled and shared across multiple tenants | Resources are pooled among members of a specific community | Resources are pooled across private and public clouds, with dynamic allocation based on workload |
| Rapid Elasticity | Elasticity may be constrained by internal resources | High elasticity with near-unlimited scalability based on demand | Elasticity exists but is constrained by the community's shared resources | Public cloud provides high elasticity, with private cloud handling more stable, predictable workloads |
| Measured Service | Internal measurement and chargeback to departments | Public provider measures and bills based on usage (e.g., | Resource usage is tracked across community members for cost-sharing | Both private and public clouds measure usage, with different |

| | | compute hours, storage) | | billing models (internal and public) |
|---|---|---|---|---|

Cloud-native applications – that are built, run, and maintained using tools, techniques and technologies for cloud computing – provide abstraction from underlying infrastructure and enhanced scalability, flexibility and reliability, which are strongest in Public and Hybrid cloud models. Cloud-native application development is driven by new software models, such as
microservices and serverless, and is made possible through technologies such as containers (i.e., Docker[21]) and container orchestration tools (i.e., Kubernetes), that are becoming the de facto leading standards for packaging, deployment, scaling and management of enterprise and business applications on cloud computing infrastructures.

Following the rise of containerization in enterprise environments, the adoption of container technologies has gained momentum in technical and scientific computing, including high-performance computing (HPC). Containers can address many HPC
problems (Mancini and Aloisio, 2015): however, security and performance overhead represent some current limits in using containerization in HPC environment (Chung et al., 2016; Abraham et al., 2020Several container platforms have been created to address the needs of the HPC community such as Shifter (Jacobsen and Canon, 2015), Singularity (Kurtzer et al., 2017) (now Apptainer), Charliecloud (Priedhorsky and Randles, 2017) and Sarus (Benedicic et al., 2019). Recently, Podman [22]has been analyzed to investigate its suitability in the context of HPC (Gantikow et al., 2020), showing some promise in bringing a
standard-based, multi-architecture enabled container engine to HPC.

## 3 Cloud Technology Landscape in Oceanography

Technological advancements in cloud computing and its foundational characteristics, services and models can provide enormous advantages for operational oceanography across the ocean architectures.

Vance et al. (2019) explored uses of the cloud for managing and analysing observational data and models workflows: for
instance, they show how cloud platforms can be supportive during the collection and the quality control of observations, reducing the risk of power outages, network connectivity or other issues related to weather conditions at sea that can compromise transmissions from sensors to the "base station". Large scale datasets related to forecast and observational oceanographic products can be stored in cloud-native storages (e.g., S3 Object Storage) and accessed from any location with public connectivity, enabling data proximate computations (Ramamurthy, 2018 This approach facilitates data-proximate
computations (Ramamurthy, 2018), allowing analysis to be performed near the data source using remote resources rather than requiring extensive local downloads and infrastructure (Zhao et al., 2015).

Nowadays, the Digital Twin of the Ocean (DTO) framework is revolutionizing ocean services, acting as a bridge between the current digitalization of processes and the future intelligence. DTO is empowering the use of advanced technologies, such as

---

[21] https://www.docker.com/
[22] https://podman.io/

Artificial Intelligence (AI) and cloud computing, for industrializing and informatizing marine sector while supporting
operations from data pooling to data processing, with final direct benefit for applications (Chen et al., 2023). It is then of
paramount importance to understand how modern computing technologies can support scientific investigation, enhance ocean
forecasting services and contribute to evolution of such systems.

To achieve this goal, analysis patterns theorized by Fowler (1997) and described for e-Science by Butler and Merati (2016),
can be applied, in a simplified way, to the ocean value chain (Alvarez Fanjul et al., 2022) explaining the added value of
adopting cloud-based solutions to improve operational forecasting workflows.

The term "analysis pattern" focuses on organizational aspects of a system since they are crucial for requirements analysis:
Geyer-Schulz and Hahsler (2001) designed a specific template for analysis patterns: starting from that and the examples
proposed by Butler and Merati (2016) for e-Science, we propose an initial analysis of Cloud Patterns (CP) for the cloud-based
OOFS processes, taking the ocean value chain components as a reference framework.

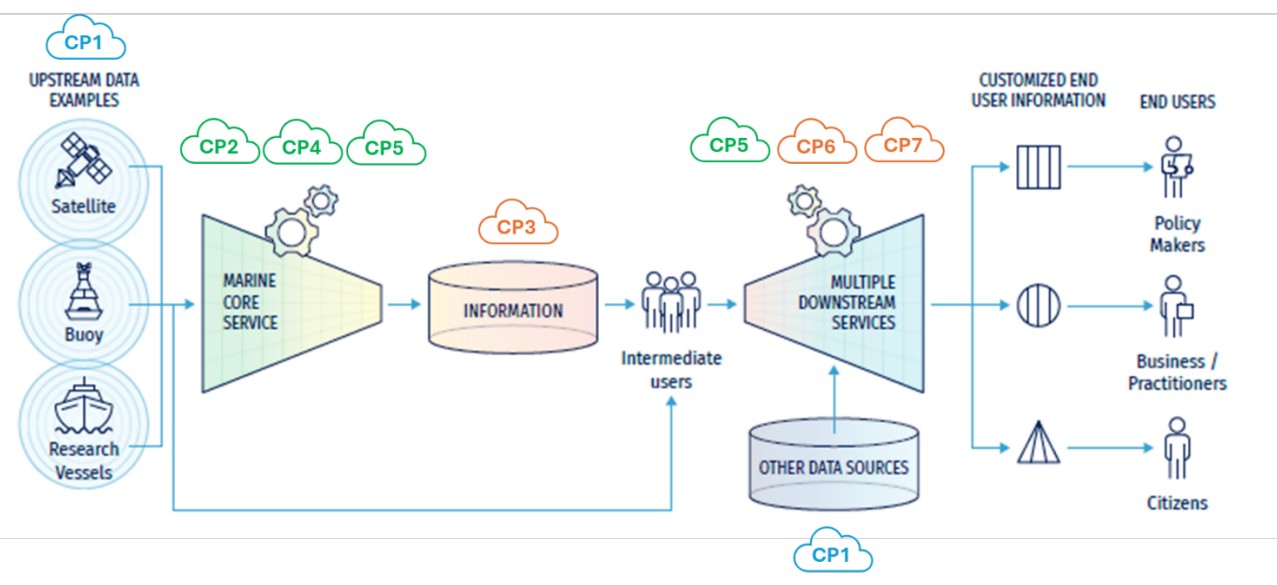

**Figure 1. The Ocean Value Chain and associated Cloud Patterns (adapted from Alvarez Fanjul et al. 2022).**

These are some initial identified cloud patterns, that are mapped in Figure 1:

- **CP1: Cloud-based management of ocean data for OOFS**. Devoted to the integration into forecasting services of
  cloud-based approach, facilitating the access to large volumes of diverse, current and authoritative data. It addresses
challenges related to locating and using large amounts of scientific data. It is particularly useful for data managers
  that needs to provide upstream data to forecasters for running one or more models, or for performing validation of
  the numerical results. In can be implemented on Hybrid/Public cloud, and the design can be based on PaaS or SaaS
  (data access as a service). It enables seamless integration of Upstream Data from multiple sources (including
  observations and forcings data used in model applications).

- **CP2. Cloud-based computing infrastructure for OOFS.** It explores cloud-based platforms and tools for running computationally intensive numerical models and procedures used for forecasting service. It benefits numerical modellers and forecasters that require high performance computing (HPC) to run a model application that can include AI/ML, pre-/post-processing. It can be implemented on Private cloud, adopting IaaS service models. It enhances the execution of the Marien Core Service by optimizing computing resources such as CPU/GPU, networking, and storage.

- **CP3. Cloud-based management of ocean data produced by OOFS.** Designed for storing and managing geospatial ocean data in the cloud, this component addresses the challenge of growing data volume with limited budgets dedicated to data management. It is valuable for data managers that need to store forecast products, including model results in native format, for further analysis and processing. Data can be stored in dedicated filesystems or databases and accessible through APIs (including GIS-based ones). It can be implemented on Private cloud, using PaaS service model. It ensures efficient storage and accessibility of data produced by the Marine Core Service, make available for dissemination to users.

- **CP4. Cloud-based computing infrastructure for OOFS disaster recovery in the Cloud.** Focused on leveraging cloud computing in the ocean forecast production pipeline to enhance robustness and meet the growing demand for scalable computational resources. It can be used by forecasters that need to OOFS on demand under unexpected situation (e.g., working as backup in case the nominal unit is down). Private/Hybrid cloud can be used, and the design can be based on PaaS or IaaS. This approach enhances the Marine Core Service by ensuring operational continuity and timely dissemination of forecast products.

- **CP5. Analysis of OOFS products in the Cloud**. Focused on performing analysis and processing of ocean data in the cloud, facilitating multi-model intercomparisons and quality assessment, even in case of larger datasets and/or on datasets from multiple sources. It is beneficial for product quality experts and data analysts in charge for quality control or for providing a private cloud-based service for pre-qualification of ocean products. It can be implemented through Hybrid/Private cloud and the design can be based on SaaS. It supports the Marine Core Service quality assurance and Downstream Services through tailored user-oriented metrics or indicators for downstream applications.

- **CP6. Visualization of OOFS in the Cloud**. Devoted to integration of cloud-based visualization capabilities to process and publish ocean products via the (cloud) service. It also addresses the need of visualizing larger amounts of data. It can be useful for data engineers and forecasters that need to create user-friendly visualizations for end-users and policy makers. It can be implemented using Private/Public cloud and the design can be based on SaaS. It supports Downstream Services by providing interactive visualization service and tailored user-oriented visual bulletins for end-users.

- **CP7. Products Dissemination and Outreach in the Cloud.** Devoted to use cloud-based platforms and tools for dissemination of OOFS products to different audiences - scientific and non-scientific. This is useful for communication experts that need to use cloud -based repository for sharing insights and digital material produced using OOFS products. It uses Hybrid/Private cloud solutions, and the design can be based on SaaS. It enhances

Multiple Downstream Services by providing customized and accessible end-users information for policy making, business, society.

Most of the challenges generically introduced in Section 2 can be still pertinent when adopting cloud computing solutions for OOFS:

- Data Security: processing oceanographic data might generate sensible information that requires proper management. In addition, downstream services might require use of data from governmental or research institutes that need to be preserved and possibly not shared.
- Costs: while cloud computing can reduce upfront infrastructure costs, it can become expensive for continuous, long-term use or for HPC tasks that require significant computational power.
- Latency and Bandwidth Limitations: ingesting or assessing large volume of ocean data on centralized cloud data centres might affect OOFS system's performances due to poor network connection.
- Dependence on Cloud Providers (Vendor Lock-In): deployment of OOFS on specific cloud providers might lead to vendor lock-in, complicating migration to another cloud provider due to proprietary technologies, APIs, or data format.
- Regulatory and Compliance Issues: cloud providers must comply with various regulatory frameworks, and using a public cloud for OOFS might complicate compliance with data protection laws or environmental regulations or even with licences.
- Limited Control over Hardware: cloud users don't have direct control over the underlying hardware, which may be a disadvantage when HPC resources need fine-tuned optimization to run OOFS.
- Impact on Code Refactoring: adapting OOFS to a cloud environment may require significant code refactoring to optimize for distributed computing, cloud-native architectures, and specific provider APIs, potentially increasing development effort and complexity.

In the following, some US and EU programmes, initiatives and projects are reported as examples on how cloud computing technologies and patterns have been used to provide services to the oceanographic and scientific community in general.

### 3.1 NOAA Open Data Dissemination & Big Data Program

NOAA's Open Data Dissemination (NODD[23]) Program is designed to facilitate public use of key environmental datasets by providing copies of NOAA's information in the Cloud, allowing users to do analyses of data and extract information without having to transfer and store these massive datasets themselves. NODD started out as the Big Data Project in April 2015 (and then later became Big Data Program); NODD currently works with three IaaS providers (Amazon Web Services (AWS), Google Cloud Platform, and Microsoft Azure) to broaden access to NOAA's data resources. These partnerships are designed to not only facilitate full and open data access at no net cost to the taxpayer but also foster innovation by bringing together the

---

[23] https://www.noaa.gov/nodd

tools necessary to make NOAA's data more readily accessible. There is over 220+ NOAA datasets on the Cloud Service Providers (CSPs) platforms. The datasets are organized by the NOAA organization who generated the original dataset (https://www.noaa.gov/nodd/datasets).

## 3.2 Copernicus Service and Data and Information Access Services

Copernicus (https://www.copernicus.eu) is the Earth Observation component of the EU Space programme, looking at the Earth
and its environment to benefit all European citizens. Copernicus is generating on a yearly basis petabyte of data and information that draw from satellite Earth Observation and in-situ (non-space) data. The up-to-date information provided by the core services (Atmosphere[24], Climate Change[25], Marine[26], Land[27], Security [28]and Emergency[29]) are free and openly accessible to users. As the data archives grow, it becomes more convenient and efficient not to download the data anymore but to analyze them where they are originally stored.

To facilitate and standardize access to data, the European Commission has funded the deployment of five cloud-based platforms (CreoDIAS[30], Mundi[31], Onda[32], Sobloo, Wekeo[33]), known as DIAS [34]– Data and Information Access Services - that provide centralized access to Copernicus data and information, as well as to processing tools. The DIAS provides users with a large choice of options to benefit from the data generated by Copernicus: to search, visualize and further process the Copernicus data and information through a fully maintained software environment while still having the possibility to download the data
to their own computing infrastructure. All DIAS platforms provide access to Copernicus Sentinel data, as well as to the information products from the six operational services of Copernicus, together with cloud-based tools (open source and/or on a pay-per-use basis). Thanks to a single access point for the entire Copernicus data and information, DIAS allows the users to develop and host their own applications in the cloud, while removing the need to download bulky files from several access points and process them locally.

## 3.3 Blue-Cloud

The European Open Science Cloud (EOSC) provides a virtual environment with open and seamless access to services for storage, management, analysis and re-use of research data, across borders and disciplines. Blue-Cloud aims at developing a

---

[24] https://atmosphere.copernicus.eu/
[25] https://climate.copernicus.eu/
[26] https://marine.copernicus.eu/
[27] https://land.copernicus.eu/en
[28] https://www.copernicus.eu/en/copernicus-services/security
[29] https://emergency.copernicus.eu/
[30] https://creodias.eu/
[31] https://mundiwebservices.com/
[32] https://www.onda-dias.eu/cms/
[33] https://www.wekeo.eu/
[34] https://www.copernicus.eu/en/access-data/dias

marine thematic EOSC to explore and demonstrate the potential of cloud-based open science for better understanding and managing the many aspects of ocean sustainability (https://blue-cloud.org/news/blue-clouds-position-paper-eosc). The Blue-Cloud platform, federating European Blue data management infrastructures (SeaDataNet[35], EurOBIS[36], Euro-Argo ERIC[37], Argo GDAC (Wong et al., 2020), EMODnet[38], ELIXIR-ENA[39], EuroBioImaging[40], Copernicus Marine, Copernicus Climate Change, and ICOS-Marine[41]) and horizontal e-infrastructures (EUDAT[42], DIAS, D4Science), provides FAIR access to multidisciplinary data, analytical tools and computing and storage facilities that support research. Blue Cloud provides Services through pilot Demonstrators for oceans, seas and freshwater bodies for ecosystems research, conservation, forecasting and innovation in the Blue Economy, and accelerates cross-discipline science, making innovative use of seamless access to multidisciplinary data, algorithms, and computing resources.

## 4 Conclusions

Cloud computing has been demonstrated to be a key driver in the digital evolution of the private sectors, offering a baseline for expanding and scaling applications and services by enhancing scalability, cost-efficiency and data processing. Service models offer different layers for pushing technological evolution, where infrastructure/platform/software can be assimilated to services that can be deployed in different cloud models, depending on the specific needs of the users in keeping resources public or private or hybrid. By leveraging on-demand computing power, big data analytics, and global data accessibility and sharing, cloud computing improves business efficiency, scientific research, and innovation, benefiting society and business. Taking these concepts as granted, cloud computing can be seen as an opportunity for operational oceanography, for enhancing ocean prediction and monitoring by exploiting its collaborative framework to support Blue Economy, sustainable ocean management and climate change mitigation actions. The simplified pattern analysis has revealed how OOFS architecture components can be implemented in cloud environment without the burden of maintaining complex infrastructure: common tasks like processing and analysing large datasets can be optimized in cloud-native storages, using software that can be integrated by AI/ML techniques for anomaly detection, or by means of specific APIs for data searching and retrieving. Cloud-based visualization and data delivery can ensure security especially for critical information that can impact decision-making, driving better-informed policies and responses in marine and coastal management.

Despite these advantages, several challenges remain, some of them partially solved with the implementation of existing deployments models (hybrid cloud, for instance): interoperability, that is one of the pillars for cloud-based environments,

---

[35] https://www.seadatanet.org/
[36] https://www.eurobis.org/
[37] https://www.euro-argo.eu/
[38] https://emodnet.ec.europa.eu/en
[39] https://elixir-europe.org/services/biodiversity
[40] https://www.eurobioimaging.eu/
[41] https://www.icos-cp.eu/observations/ocean/otc
[42] https://www.eudat.eu/

requires definition of data standards and adoption of best practices. Security in data access/sharing as well as costs associated

with running of forecasting systems can raise constraints for vendor lock-in and long-term sustainability.

Promoting a collaborative framework among existing and new centers could be seen as one promising approach for fostering innovation, collaboration and more efficient ocean prediction and monitoring: by leveraging shared cloud-based resources, forecasting centres can combine their expertise and share data and tools, supporting the creation of a "digital twin" of the ocean, to use for wide range of applications for managing and protecting our ocean.

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

## Competing interests

The contact author has declared that none of the authors has any competing interests.

## Authors contribution

SC contributed to the conceptualization, writing and validation. GP contributed to the writing and validation.

**Acknowledgements**

The Authors are grateful to Dr. Marco Mancini for sharing his expert view on Cloud Computing technology, and to Dr. Enrique Alvarez Fanjul, for analysing Section 3 robustness sharing his expert view on the impact of using Cloud Computing in support to Operational Oceanography. They also extend their gratitude to the reviewers, Dr. Alvaro Lorenzo Lopez and Dr. Miguel Charcos-Llorens, and the to the topic editor, Dr. Jay Pearlman, for their valuable and insightful comments that significantly improved this manuscript.

405