# Peer review of "Distributed Environments for Ocean Forecasting: the role of Cloud Computing"

_State of the Planet, 2024_

## Referee Comment (RC1)

**Line 25**

McCarthy envisioned a future where users would be able to access computing resources over a network, paying for usage just like utilities. This concept did not specifically refer to cloud computing as we know it today (i.e., a scalable, on-demand infrastructure of virtualized resources). Instead, his idea revolved around time-sharing systems, where multiple users could share a single powerful computer. At that time, the term "cloud" did not exist, and computing was centralized, not distributed over many interconnected networks like the Internet today.

**Line 29:**

It is true that Amazon was one of the first companies to offer true cloud infrastructure services at scale with Amazon Web Services (AWS), starting with Amazon S3 in 2006 and followed by EC2. These services allowed businesses to rent storage and computing power, marking a significant shift in how cloud computing was deployed and understood. They made it easy for organizations to scale resources based on need, which contributed to the mass adoption of cloud technologies. However, they did not invent the concept of cloud computing. In the 90s, grid computing already referred to distributed computing resources to work on large tasks. Although task-specific, it is a similar concept as cloud computing. Some companies already provided Application Service Providers (ASPs), providing software applications to businesses over the Internet, an early version of delivering services remotely over a network. Additionally, Salesforce offered in 1999 an early version of the Software-as-a-Service (SaaS), model offered by Amazon. Concerning the term, it was neither new. In 1996, the term "cloud computing" was used by Compaq that described the future of computing services being provided through the web. It was also used in the telecommunications industry in the early 2000s, but it is not relevant since it referred to network infrastructure using virtual private networks (VPNs) - just a note.

In brief, the term "cloud computing" did not originate with Amazon in 2006, but Amazon was instrumental in popularizing the term and the modern concept of cloud infrastructure with AWS. The paper could be adjusted to say that Amazon popularized the modern form of cloud computing with its AWS offerings rather than claiming that Amazon coined the term or was the first to provide cloud services.

**Table1**
- Broad network access: Do you mean standard mechanisms such as HTTP/HTTPS or APIs? Please clarify.
- Definitions are very unbalanced and unclear. I would prefer to see some comparison of the various services such as: (just as an example)

| Characteristics | Primary Focus | Client Perspective | Cloud Provider Perspective | Example |
|---|---|---|---|---|
| On-Demand Self-Service | Users can provision | Users can request and | Automatically provide | A developer launches a |

| | computing resources (e.g., storage, VMs) automatically, without requiring human interaction with the service provider | configure resources like virtual machines, storage, or applications when needed, directly from a web interface or API | resources in response to user requests without manual intervention | virtual machine on a cloud platform using a dashboard or API in minutes, without needing to contact support |
|---|---|---|---|---|
| Broad Network Access | Cloud resources are available over a network and accessible through standard mechanisms from various devices | Users can access cloud services from a range of devices (e.g., phones, laptops) through standard protocols like HTTP/HTTPS | Ensure cloud services can be accessed consistently and securely from different client devices | A user edits a document stored in the cloud from a laptop at home, and then continues editing from a smartphone while commuting |
| Resource Pooling | Cloud providers pool resources to serve multiple users (tenants) dynamically, with no fixed assignment to any one user | Users don't know the exact physical location of the resources they are using, but they get what they need as required | Dynamically allocate physical and virtual resources across many customers to maximize efficiency and utilization | Multiple customers use the same set of servers and storage, but their workloads are isolated through virtualization technologies |
| Rapid Elasticity | Cloud resources can be quickly scaled up or down to meet demand, often appearing limitless to the user | Users can automatically scale their resources up or down based on their needs, without delays | Automatically add or remove resources in response to changing demand, ensuring that the user has sufficient capacity | An e-commerce website automatically scales up its computing resources during a flash sale, then scales down when the traffic subsides |
| Measured Service | Cloud systems automatically control and optimize resource usage by tracking it and charging based on actual consumption | Users only pay for the amount of resources (e.g., storage, CPU, bandwidth) they actually use, with detailed reporting | Track resource consumption at various levels (e.g., storage, CPU usage) and optimize based on real-time monitoring | A company receives a monthly bill detailing how much computing power and storage they used, ensuring that they are |

| | | | | billed accurately based on consumption |
|---|---|---|---|---|
| | | | | |

**Table 3:**

For Community cloud it is worth mentioning D4Science that is used for Blue Cloud which is the node of oceanography of EOSC. Not sure if the EGI and/or D4Science follow in fact hybrid models instead. It is worth confirming this.

**Explanation of Table 1, 2 and 3 disconnected**

I am missing the connection between the definitions of the five essential characteristics, three service models, and four deployment models. For example, how deployment models influence the service models. I believe that private Cloud more often use IaaS or PaaS, where organizations need control over sensitive data and compliance but want to maintain cloud-like services internally. Public Cloud is ideal for all service models—IaaS, PaaS, SaaS, and FaaS—as it provides high elasticity, scalability, and a broad range of services with minimal infrastructure management from users. As for private cloud, community Cloud typically uses IaaS or PaaS, where several organizations collaborate on shared resources, often due to compliance or regulatory needs. Finally, hybrid cloud: Combines IaaS, PaaS, and SaaS elements, often using public clouds for flexible, scalable services and private clouds for sensitive or mission-critical data and applications.

A table could be also added to explain how characteristics are managed in the various deployment models. Just as an example to be improved:

| Characteristic | Private Cloud | Public Cloud | Community Cloud | Hybrid Cloud |
|---|---|---|---|---|
| On-Demand Self-Service | Managed internally, self-service for internal teams | Users provision services via public provider's API or portal | Self-service for community members, often through secure portals | Self-service across both public and private clouds, with potential for complex management |
| Broad Network Access | Limited to internal users or authorized external users (VPN, private network) | Accessible over the public internet via standard protocols (e.g., HTTP) | Restricted to community members with specific access | Accessible over both public and private networks, often with encrypted or dedicated connections |

| Resource Pooling | Resources are pooled internally for organizational needs | Resources are pooled and shared across multiple tenants | Resources are pooled among members of a specific community | Resources are pooled across private and public clouds, with dynamic allocation based on workload |
|---|---|---|---|---|
| Rapid Elasticity | Elasticity may be constrained by internal resources | High elasticity with near-unlimited scalability based on demand | Elasticity exists but is constrained by the community's shared resources | Public cloud provides high elasticity, with private cloud handling more stable, predictable workloads |
| Measured Service | Internal measurement and chargeback to departments | Public provider measures and bills based on usage (e.g., compute hours, storage) | Resource usage is tracked across community members for cost-sharing | Both private and public clouds measure usage, with different billing models (internal and public) |

**Line 60**: The explanation emphasizes using multiple public cloud providers, but multi-cloud strategies can also include private clouds. Some organizations use a mix of private and public clouds as part of their multi-cloud setup. You may want to acknowledge that multi-cloud isn't just limited to public providers. Moreover, while the explanation focuses on the benefits, it might be useful to briefly mention some of the challenges of multi-cloud computing, such as increased complexity in management (having to manage multiple cloud environments), integration issues (ensuring compatibility between services across clouds), and security concerns (handling security policies across different providers).

**Line 65**: distributed cloud-edge computing also helps reduce bandwidth usage and improve data privacy by processing data locally. I believe that EGI also has some of the distributed cloud-edge aspects (to be confirmed).

**Line 78**: It's not the infrastructure itself that is "agile," but rather the processes (Agile and DevOps) that benefit from the scalability and automation cloud infrastructure provides.

**Line 81 & 84**:Instead of "Linux containers," it's more accurate to refer to them as containers or Docker containers. You could mention that Docker originally leveraged Linux-based containers, but today it supports other OS environments as well. That is one of the strong advantages of this technology.

**Line 85**: While HPC is a form of technical and scientific computing, a small rephrasing can help clarify this category without creating confusion. It is not the only one.

**Line 94**: Vance reference is from 2016. Since these technologies are evolving rapidly it would be interesting to mention a more up to date publication about this topic. Additionally, you present the advantages of cloud computing in this paragraph but it is worth to mention the down sides as well:

- Data Security and Privacy Concerns: Storing and processing large amounts of oceanographic data in the cloud raises security and privacy concerns, especially when dealing with sensitive information or data from governmental or research institutions.
- Costs for Long-Term or High-Performance Usage: While cloud computing can reduce upfront infrastructure costs, it can become expensive for continuous, long-term use or for high-performance computing (HPC) tasks that require significant computational power.
- Latency and Bandwidth Limitations: Cloud computing depends on network connections, and latency or bandwidth limitations could affect real-time processing, especially if data is being sent from remote ocean observation platforms to centralized cloud data centers.
- Dependence on Cloud Providers (Vendor Lock-In): Relying heavily on a specific cloud provider can lead to vendor lock-in, where migrating to another cloud provider becomes difficult or expensive due to proprietary technologies, APIs, or data formats.
- Regulatory and Compliance Issues: Cloud providers must comply with various regulatory frameworks, and using a public cloud for operational ocean forecasting may complicate compliance with data protection laws or environmental regulations.
- Limited Control over Hardware: Cloud users don't have direct control over the underlying hardware, which may be a disadvantage when high-performance computing (HPC) resources need fine-tuned optimization for ocean forecasting models.

---

## Author Comment (AC3)

**Title: Distributed Environments for Ocean Forecasting: the role of Cloud Computing**

**Authors: S. Ciliberti and G. Coro**

**MS No.: sp-2024-37**

**Report: Ocean prediction: present status and state of the art**

Dear Reviewer,

We would warmly thank you for the detailed and fruitful critical analysis of the paper, and for having provided many items to support further improvement of the overall manuscript. We carefully analysed them and in the following we provide punctual answers to your questions/remarks proposing new drafted paragraphs.

In the following:

- In black, your original comments.
- In blue, our answers to them.

**General comments**

The paper is well-organized, with distinct sections explaining cloud computing concepts, service models, deployment models, and their relevance to Operational Ocean Forecasting. It provides a comprehensive overview of cloud computing, including its essential characteristics and how they apply to scientific and operational oceanography. The inclusion of real-world examples, such as NOAA's Open Data Dissemination Program and the Copernicus Service, adds value by grounding theoretical discussions in practical applications. Additionally, it mentions important technologies for cloud-native development, such as Docker, Kubernetes, and HPC-focused container platforms which are crucial in scientific computing environments.

Nevertheless, there are some areas of improvement.

The paper focuses heavily on the benefits of cloud computing, but as I will discuss in my comments, it would be valuable to address some downsides or challenges. For instance, data security, cost management, and performance issues in high-performance computing (HPC) settings should be mentioned, as these are relevant concerns for any organization or scientific body adopting cloud solutions. There are some lacks of technical precisions that I also comment below.

Suggested concepts have been incorporated in the paper: a revised version of the HPC section is proposed in our new revised version of the manuscript, focusing on benefits and challenges. Also, punctual answers to your questions in the next sections.

The paper could benefit from more discussion on emerging trends like AI/ML integration with cloud computing in operational oceanography, which is becoming increasingly important for predictive models and real-time analytics. In this sense, there are some initiatives happening using EGI such as the iMagine project.

In our revised version of the manuscript, we cite the paper included also in this Special Issue: https://sp.copernicus.org/preprints/sp-2024-18/ .

You could also point out the collaborative aspect of cloud computing. Many cloud-based projects in oceanography, like Copernicus and NOAA, focus on data sharing and collaborative research via Virtual Research Environments. Cloud platforms enable large-scale collaborative environments where multiple stakeholders can work with shared datasets and tools, thus improving international collaboration and research outcomes.

The revised version of the manuscript improves the discussion on collaborative cloud-edge computing.

Another topic I am missing in this paper is data interoperability and FAIR principles. A key challenge in oceanography is ensuring that large, distributed datasets follow FAIR principles (Findable, Accessible, Interoperable, Reusable). Cloud computing can enhance data interoperability through standardization of formats, APIs, and access protocols, ensuring that datasets can be easily shared, accessed, and reused by researchers globally.

The manuscript has been improved to include a discussion on the FAIR principles and how cloud computing supports it. Following your suggestion, we discussed also the challenges in operational ocean forecasting, proposing a) a new dedicated section that talks about cloud computing in operational oceanography and b) a table that analyses the benefits in adopting cloud computing for developing the fundamental components of an operational forecasting systems, including challenges (as proposed later).

The tables 1, 2, 3 and 4 provide valuable definitions, models and patterns. The main text could do a better job of integrating and leveraging these definitions more effectively to support the overall analysis. The main text mentions these characteristics but doesn't always connect them directly to specific use cases in ocean forecasting or other scientific applications. For example, when discussing data storage and management, it could explicitly reference the measured service characteristic to highlight how cloud providers charge based on resource usage. Similarly, the broad network access characteristic could be tied to how cloud services enable global access to oceanographic data from remote locations.

We totally reorganize the section that discusses foundational concepts of cloud computing. More specific answers will be provided in the next section, that acquire your suggestions in revising the tables.

Including an overall landscape of cloud technologies in oceanography would add significant value to the paper. It would provide readers with a broader view of how various cloud technologies are applied across the oceanographic field, helping them understand the diversity of tools, platforms, and strategies currently in use. In fact, a landscape analysis would give readers a complete picture of the range of cloud-based tools, platforms, and applications being used for different tasks in oceanography. This could include everything from data collection and storage to forecasting models, visualization, and collaborative platforms. By presenting a landscape, you can highlight emerging trends (as I suggested earlier) in the field, such as the growing use of AI and machine learning, edge computing, and serverless technologies. This would position the paper as forward-thinking and relevant to ongoing technological advances. You could include a dedicated section (e.g. "Cloud Technology Landscape in Oceanography"), summarizing the technologies used at each stage of oceanographic data collection, analysis, and dissemination. Visual aids such as diagrams or tables could map out which cloud platforms are used for different tasks, helping readers see how different cloud technologies fit into the broader landscape of oceanography.

The new proposed dedicated section on cloud computing and operational oceanograpjy has been introduced in this revised version of the manuscript with suggested title.

**Specific comments**

*Line 25*

McCarthy envisioned a future where users would be able to access computing resources over a network, paying for usage just like utilities. This concept did not specifically refer to cloud computing as we know it today (i.e., a scalable, on-demand infrastructure of virtualized resources). Instead, his idea revolved around time-sharing systems, where multiple users could share a single powerful computer. At that time, the term "cloud" did not exist, and computing was centralized, not distributed over many interconnected networks like the Internet today.

Thanks for this comment. Indeed, the original version lacked in presenting the most relevant milestones in cloud computing. The paragraph has been rephrased, precising that McCarthy's vision can be seen mostly as a sorth of metaphor for the Internet.

Please find in the following the new paragraph:

[…] The term originated as a metaphor for the Internet which is, in essence, a network of networks providing remote access to a set of decentralized IT resources. In the early 1960s, J. McCarthy introduced the concept of computing as Utility: "If computers of the kind I have advocated become the computers of the future, then computing may someday be organized as a public utility just as the telephone system is a public utility.… The computer utility could become the basis of a new and important industry". This idea opened to the concept of having services on Internet so users could benefit of them for their applications.

This brief overview on history of Cloud Computing has been enriched by citing Licklider and Chellappa: the first envisioned a world where interconnected systems of computers could communicate; the second who introduced for the fist time the term "cloud computing".

*Line 29*

It is true that Amazon was one of the first companies to offer true cloud infrastructure services at scale with Amazon Web Services (AWS), starting with Amazon S3 in 2006 and followed by EC2. These services allowed businesses to rent storage and computing power, marking a significant shift in how cloud computing was deployed and understood. They made it easy for organizations to scale resources based on need, which contributed to the mass adoption of cloud technologies. However, they did not invent the concept of cloud computing. In the 90s, grid computing already referred to distributed computing resources to work on large tasks. Although task-specific, it is a similar concept as cloud computing. Some companies already provided Application Service Providers (ASPs), providing software applications to businesses over the Internet, an early version of delivering services remotely over a network. Additionally, Salesforce offered in 1999 an early version of the Software-as-a-Service (SaaS), model offered by Amazon. Concerning the term, it was neither new. In 1996, the term "cloud computing" was used by Compaq that described the future of computing services being provided through the web. It was also used in the telecommunications industry in the early 2000s, but it is not relevant since it referred to network infrastructure using virtual private networks (VPNs) - just a note.

In brief, the term "cloud computing" did not originate with Amazon in 2006, but Amazon was instrumental in popularizing the term and the modern concept of cloud infrastructure with AWS. The paper could be adjusted to say that Amazon popularized the modern form of cloud computing with its AWS offerings rather than claiming that Amazon coined the term or was the first to provide cloud services.

Linked to previous comment, the history of cloud computing's term has been revised to align to your suggestion.

The new paragraph then looks like this in the following:

"[…] . In the same period, J. C. R. Licklider envisioned a world where interconnected systems of computers could communicate and inter-operate: that was the milestone of the modern cloud computing. In the late 1990s, R. Chellappa introduced for the first time the term "cloud computing" as a new computing paradigm (Chellappa, 1997), "where the boundaries of computing will be determined by economic rationale rather than technical limits alone", dealing with concepts such as expandable and allocatable resources that can ensure cost-efficiency, scalability, and business value. In the same period, Compaq Computer Corporation adopted the concept of "cloud" in its business plan, as term for evolving the technological capacity of the company itself in offering new scalable and expandable services to customers over the Internet".

The remaining paragraph has been kept as in its original format, more focused on recent achievements (Amazon, Google, Micosoft…).

*Table1*

Broad network access: Do you mean standard mechanisms such as HTTP/HTTPS or APIs? Please clarify.

We could consider both cases: HTTP/HTTPS for instance can be used to access applications related to visualization, while APIs can be used to access data.

Definitions are very unbalanced and unclear.

I would prefer to see some comparison of the various services such as: (just as an example)

| Characteristics | Primary Focus | Client Perspective | Cloud Provider Perspective | Example |
|---|---|---|---|---|
| On-Demand Self-Service | Users can provision computing resources (e.g., storage, VMs) automatically, without requiring | Users can request and configure resources like virtual machines, storage, or applications when | Automatically provide resources in response to user requests without manual intervention | A developer launches a virtual machine on a cloud platform using a dashboard or API in minutes, without |

| | | | | |
|---|---|---|---|---|
| | human interaction with the service provider | needed, directly from a web interface or API | | needing to contact support |
| Broad Network Access | Cloud resources are available over a network and accessible through standard mechanisms from various devices | Users can access cloud services from a range of devices (e.g., phones, laptops) through standard protocols like HTTP/HTTPS | Ensure cloud services can be accessed consistently and securely from different client devices | A user edits a document stored in the cloud from a laptop at home, and then continues editing from a smartphone while commuting |
| Resource Pooling | Cloud providers pool resources to serve multiple users (tenants) dynamically, with no fixed assignment to any one user | Users don't know the exact physical location of the resources they are using, but they get what they need as required | Dynamically allocate physical and virtual resources across many customers to maximize efficiency and utilization | Multiple customers use the same set of servers and storage, but their workloads are isolated through virtualization technologies |
| Rapid Elasticity | Cloud resources can be quickly scaled up or down to meet demand, often appearing limitless to the user | Users can automatically scale their resources up or down based on their needs, without delays | Automatically add or remove resources in response to changing demand, ensuring that the user has sufficient capacity | An e-commerce website automatically scales up its computing resources during a flash sale, then scales down when the traffic subsides |
| Measured Service | Cloud systems automatically control and optimize resource usage by tracking it and charging based on actual consumption | Users only pay for the amount of resources (e.g., storage, CPU, bandwidth) they actually use, with detailed reporting | Track resource consumption at various levels (e.g., storage, CPU usage) and optimize based on real-time monitoring | A company receives a monthly bill detailing how much computing power and storage they used, ensuring that they are billed accurately based on consumption |

Your proposed table has been incorporated in the manuscript and indeed it gives now more clear overview of the different perspectives of cloud computing Essential Characteristics, completed by provided examples. We thank you for this idea.

Similarly to Table 1, we would like to propose a revised version of Table 2 related to Service Models. Similarly to what done for Essential Characteristics, Table 2 would show the 2 different perspectives – client/providers – in adopting one of the proposed Service Models as specified by NIST, completed with Use Cases as examples. In the following, the proposed table.

| Service Model | Primary Focus (from Mell and Grance, 2011) | Client Perspective | Provider Perspective | Use Cases |
|---|---|---|---|---|
| Infrastructure as a Service (IaaS) | The capability provided to the consumer is to | Renting and managing computing | Provisioning of computing resources in a | Suitable for organizations that want full control over |

| | | | | |
|---|---|---|---|---|
| | provision processing, storage, networks, and other fundamental computing resources where the consumer can deploy and run arbitrary software. | resources in a virtualized infrastructure. | virtualized infrastructure. | their infrastructure resources (virtual machines, networks, storage) that want their flexibility in customizing software stack and applications, including data processing and backup.Examples: Amazon EC2, Microsoft Azure, etc. |
| Platform as a Service (PaaS) | The capability provided to the consumer is to deploy onto the cloud infrastructure consumer-created or acquired applications created using programming languages, libraries, services, and tools supported by the provider. | Easing applications deployment without taking care of the infrastructure and middleware. Dependency on provider's platform. | Provisioning and management of the platform. | Suitable for developers and organizations that want to develop, deploy and maintain applications without the burden of managing the underlying infrastructure (virtual machines, network and storage), that is provisioned and deployed by the providers. Examples: Google App Engine, Microsoft Azure App Services, etc. |
| Software as a Service (SaaS) | The capability provided to the consumer is to use the provider's applications running on a cloud infrastructure. | Using directly sorftware applications via Internet (e.g., web browser or using a client), decreasing costs related to licences. | Provisioning and management of the software applications, including customer support. | It enables organizations to focus on their core business activities while relying on the expertise and infrastructure provided by the SaaS provider. Examples: Google Drive, Dropbox, Microsoft 365. |

*Table 3*

For Community cloud it is worth mentioning D4Science that is used for Blue Cloud which is the node of oceanography of EOSC. Not sure if the EGI and/or D4Science follow in fact hybrid models instead. It is worth confirming this.

We verified and we confirm that both EGI and D4Science adopt hybrid models. They are now both included in Table 3 under the Community Cloud row.

*Explanation of Table 1, 2 and 3 disconnected*

To better link the concepts, the whole section has been reorganized, providing a description of the cloud computing concepts.

I am missing the connection between the definitions of the five essential characteristics, three service models, and four deployment models. For example, how deployment models influence the service models. I believe that private Cloud more often use IaaS or PaaS, where organizations need control over sensitive data and compliance but want to maintain cloud-like services internally. Public Cloud is ideal for all service models—IaaS, PaaS, SaaS, and FaaS—as it provides high elasticity, scalability, and a broad range of services with minimal infrastructure management from users. As for private cloud, community Cloud typically uses IaaS or PaaS, where several

organizations collaborate on shared resources, often due to compliance or regulatory needs. Finally, hybrid cloud: Combines IaaS, PaaS, and SaaS elements, often using public clouds for flexible, scalable services and private clouds for sensitive or mission-critical data and applications.

A table could be also added to explain how characteristics are managed in the various deployment models. Just as an example to be improved:

| Characteristic | Private Cloud | Public Cloud | Community Cloud | Hybrid Cloud |
|---|---|---|---|---|
| On-Demand Self-Service | Managed internally, self-service for internal teams | Users provision services via public provider's API or portal | Self-service for community members, often through secure portals | Self-service across both public and private clouds, with potential for complex management |
| Broad Network Access | Limited to internal users or authorized external users (VPN, private network) | Accessible over the public internet via standard protocols (e.g., HTTP) | Restricted to community members with specific access | Accessible over both public and private networks, often with encrypted or dedicated connections |
| Resource Pooling | Resources are pooled internally for organizational needs | Resources are pooled and shared across multiple tenants | Resources are pooled among members of a specific community | Resources are pooled across private and public clouds, with dynamic allocation based on workload |
| Rapid Elasticity | Elasticity may be constrained by internal resources | High elasticity with near-unlimited scalability based on demand | Elasticity exists but is constrained by the community's shared resources | Public cloud provides high elasticity, with private cloud handling more stable, predictable workloads |
| Measured Service | Internal measurement and chargeback to departments | Public provider measures and bills based on usage (e.g., compute hours, storage) | Resource usage is tracked across community members for cost-sharing | Both private and public clouds measure usage, with different billing models (internal and public) |

A revised version of the table you proposed has been included in the manuscript. We thank you for the suggestion.

*Line 60*

The explanation emphasizes using multiple public cloud providers, but multi-cloud strategies can also include private clouds. Some organizations use a mix of private and public clouds as part of their multi-cloud setup. You may want to acknowledge that multi-cloud isn't just limited to public providers. Moreover, while the explanation focuses on the benefits, it might be useful to briefly mention some of the challenges of multi-cloud computing, such as increased complexity in management (having to manage multiple cloud environments), integration issues (ensuring compatibility between services across clouds), and security concerns (handling security policies across different providers).

The paragraph on multi-cloud has been revised according to your comments, highlighting that the technology can include private, public or a combination public-private clouds: "Multi-cloud computing refers to the strategy of using multiple cloud service providers, allowing organizations leveraging the services of two or more public/private cloud providers or a combination public-private, combining their offerings to build and manage their applications and infrastructure". Additionally, a brief discussion on the challenges has been included.

A new paragraph on edge computing has been added to complete the overview on available cloud strategies and to link to next part related to distributed cloud-edge computing.

*Line 65*

Distributed cloud-edge computing also helps reduce bandwidth usage and improve data privacy by processing data locally. I believe that EGI also has some of the distributed cloud-edge aspects (to be confirmed).

The comment has been incorporated into the paragraph dedicated to distributed computing.

*Line 78*

It's not the infrastructure itself that is "agile," but rather the processes (Agile and DevOps) that benefit from the scalability and automation cloud infrastructure provides.

The reference to the "agile" has been removed: indeed we agree it was not pertinent.

*Line 81 & 84*

Instead of "Linux containers," it's more accurate to refer to them as containers or Docker containers. You could mention that Docker originally leveraged Linux-based containers, but today it supports other OS environments as well. That is one of the strong advantages of this technology.

We agree on your comment and we refer generically to containers: "Cloud-native application development is driven by new software models, such as microservices and serverless, and is made possible through technologies such as containers (i.e., Docker) and container orchestration tools (i.e., Kubernetes), that are becoming the de facto leading standards for packaging, deployment, scaling and management of enterprise and business applications on cloud computing infrastructures.

*Line 85*

While HPC is a form of technical and scientific computing, a small rephrasing can help clarify this category without creating confusion. It is not the only one.

The paragraph has been revised, including references on current challenges in adopting containers in HPC environment.

*Line 94*

Vance reference is from 2016. Since these technologies are evolving rapidly it would be interesting to mention a more up to date publication about this topic. Additionally, you present the advantages of cloud computing in this paragraph but it is worth to mention the down sides as well:

- Data Security and Privacy Concerns: Storing and processing large amounts of oceanographic data in the cloud raises security and privacy concerns, especially when dealing with sensitive information or data from governmental or research institutions.
- Costs for Long-Term or High-Performance Usage: While cloud computing can reduce upfront infrastructure costs, it can become expensive for continuous, long-term use or for high-performance computing (HPC) tasks that require significant computational power.
- Latency and Bandwidth Limitations: Cloud computing depends on network connections, and latency or bandwidth limitations could affect real-time processing, especially if data is being sent from remote ocean observation platforms to centralized cloud data centers.
- Dependence on Cloud Providers (Vendor Lock-In): Relying heavily on a specific cloud provider can lead to vendor lock-in, where migrating to another cloud provider becomes difficult or expensive due to proprietary technologies, APIs, or data formats.
- Regulatory and Compliance Issues: Cloud providers must comply with various regulatory frameworks, and using a public cloud for operational ocean forecasting may complicate compliance with data protection laws or environmental regulations.

- Limited Control over Hardware: Cloud users don't have direct control over the underlying hardware, which may be a disadvantage when high-performance computing (HPC) resources need fine-tuned optimization for ocean forecasting models.

This section has been deeply revised, giving more emphasis on the possibility offered by cloud computing to improve specific components of the Operational Ocean Forecasting Systems. Additionally, the list of challenges has been included and slightly modified.

---

## Author Comment (AC4)

**Title: Distributed Environments for Ocean Forecasting: the role of Cloud Computing**

**Authors: S. Ciliberti and G. Coro**

**MS No.: sp-2024-37**

**Report: Ocean prediction: present status and state of the art**

Dear Reviewer,

We would warmly thank you for the detailed and fruitful critical analysis of the paper, and for having provided many items to support further improvement of the overall manuscript. We carefully analysed them and in the following we provide punctual answers to your questions/remarks proposing new drafted paragraphs.

In the following:

- In black, your original comments.
- In blue, our answers to them.

**General comments**

The document is very well written, with very clear English that it is easy to follow and understand.

The manuscript presents a basic state of the art of the cloud concept and cloud providers angled to modelers to inform them with the advantages of running their models in a cloud environment.

The manuscript contains enough relevant references.

My only concern is the manuscript is not particularly innovative or exciting; presents cloud concepts that have been used for almost two decades. I recognize that the target audience may not be familiar with those concepts, and there is nothing wrong with presenting them again if they lead somewhere. And that is the biggest problem with the manuscript; while I see where the authors are leading the reader, there is no enough strong arguments in section 2 to properly inform or convince a modeler that the cloud approach is the right one.

Considering your main concern, we totally revised the structure of the paper, providing in Section 2 basic key concepts on Cloud Computing and in Section 3 an outlook on possible benefits/challenges of such technologies in operational ocean forecasting systems.

I am not against the publication of this work, but I would like to ask the authors to revise their work;

- The introduction should introduce the problem they are trying to solve, not just the concept of the cloud. The content of the current introduction is relevant but reading it I don't understand the problem the paper tries to address.

A new version of the introduction is now proposed, trying to better address the scope of the manuscript and the relevance of considering cloud computing for improving ocean value chain.

- Section 1 is fine, informative. The authors talk about Linux containers in page 6, but I believe they are trying to describe container technology, which is not just used for Linux, they can be used for any OS. Linux containers is an umbrella term used for container technologies under Linux. Please clarify this potentially swapping Linux containers for just containers. In the same page the authors claim that Docker has not made strides into the HPC world due to technical limitations. There is no text or reference to substantiate such claim.

The reference originally provided to Linux-containers has been removed and the section has been restructured to a) introduce brief history on cloud computing and b) discussing cloud computing technology through a set of tables, elaborated from the additional provided comments by Reviewer #1. The section has been revised, the message on technical limitations in using containers in HPC environments is now supported by references and revised.

- Section 2. This should be the main part of the manuscript where the authors should work more, better articulating how modelers can leverage cloud technologies. The two examples provided are relevant but reading them they are just informative, there is no clear narrative giving the reader a cohesive view. This could be address with some extra text after the three examples.

The section has been totally rewritten, giving an overview of how cloud computing technology can support specific components of the ocean value chain.

There is no closing section; what have we learnt?, what are the future cloud technological developments and trends the reader should keep an eye on?.

"Conclusions" have been included now.

I am more than happy to provide the authors with further comments if they have any questions.

We would be more than happy to have new feedback from you based on new revised and (we hope) improved version of the manuscript.